# Avoidant and Approach-Oriented Coping Strategies, Meaning Making, and Mental Health Among Adults Bereaved by Suicide and Fatal Overdose: A Prospective Path Analysis

**DOI:** 10.3390/bs15050671

**Published:** 2025-05-14

**Authors:** Jamison S. Bottomley, Robert A. Neimeyer

**Affiliations:** 1National Crime Victims Research and Treatment Center, Department of Psychiatry and Behavioral Sciences, Medical University of South Carolina, 67 President Street, Charleston, SC 29425, USA; 2Portland Institute for Loss and Transition, Portland, OR 97223, USA; neimeyer@portlandinstitute.org

**Keywords:** suicide loss, overdose, prolonged grief, posttraumatic stress, depression, bereavement, avoidance, approach, meaning making, mediation

## Abstract

Adults bereaved by the suicide or overdose death of someone close to them are vulnerable to adverse mental health outcomes, but little is known about how these individuals utilize avoidance- and approach-based coping strategies, how these strategies relate to outcomes, and what accounts for these associations. Informed by contemporary theories of bereavement, we utilize prospective data from suicide- and overdose-bereaved adults (*N* = 212) who completed two waves of online data collection approximately two years following the death (T1 and T2; six months apart) to examine the mediating role of meaning making in the relationship between coping strategies and grief-related mental health outcomes, such as prolonged grief (PG), posttraumatic stress (PTS), and depression. Path analysis with mediation was used to investigate the relations between coping strategies at T1, meaning making at T2, and mental health outcomes at T2. The results indicated direct effects of avoidant coping at T1 in predicting higher PG and PTS symptoms at T2, while approach-based coping at T1 indirectly predicted an improvement in all three T2 outcomes due to increased meaning making. These results suggest that avoidance-based strategies directly and indirectly contribute to poorer outcomes and impaired meaning making processes, while approach-based strategies lead to increased meaning making and adaptation to loss among suicide and overdose loss survivors. The clinical implications and future directions for research are discussed.

## 1. Introduction

Deaths due to suicide and fatal overdoses continue to mount in the United States. Recent estimates suggest that well over 100,000 drug overdose fatalities ([1]) and 50,000 suicide deaths occur in the United States each year ([11]). These figures suggest an exorbitant number of persons bereaved by the fatal overdose or suicide of someone close to them in the United States who are at risk for a number of mental health challenges, such as elevated levels of prolonged grief (PG), posttraumatic stress (PTS), and depressive symptoms ([13]; [29]; [48]), among other difficulties ([15]). Given the profound levels of mental health sequalae and their associated impairment among the overdose- and suicide-bereaved, it is important to investigate the ways in which survivors of such losses engage in specific coping strategies and how these strategies relate to mental health outcomes over time.

Coping behaviors have been categorized in a variety of ways, but the consensus suggests that coping strategies employed following highly stressful events, such as interpersonal loss, fall into three categories: problem-focused coping, emotion-focused coping, and avoidant coping. Problem-focused coping involves planning or engaging in activities to overcome challenges that may be causing distress (e.g., information gathering, positive reframing), while emotion-focused coping includes a range of strategies aimed to regulate one’s emotional reactions to the stressful incident (e.g., acceptance, seeking emotional support; [31]). Avoidant coping includes efforts to distract oneself or disengage from sources of affective distress (e.g., behavioral disengagement, self-distraction). Evidence suggests that some forms of coping may be more advantageous than others, but with inconsistent results. For instance, while some studies of trauma-exposed individuals have found that problem-focused strategies were associated with reduced PTS symptoms ([26]), others have found the opposite to be true ([53]). Other research has documented that coping behaviors that directly confront emotional distress or help the individual engage in meaningful activities are associated with lower depressive symptoms, while avoidant strategies that promote distraction or disengagement have been associated with greater depressive ([37]) and PTS symptoms ([28]). Although avoidant coping has been associated with PG symptoms following loss ([4]), the relationship between problem- or emotion-focused strategies and other grief-related outcomes (PTS, depression) is less clear ([50]), particularly in the context of suicide and overdose bereavement. Moreover, although prior research has suggested relationships between coping strategies and mental health outcomes, little is known about *how* these strategies may relate to outcomes. Clarifying the association between specific coping strategies and mental health outcomes, including the identification of possible mechanisms that may explain the association, is important and can inform clinical interventions that target such underlying processes.

Based on contemporary bereavement theory, which recognizes the importance of making sense of the death ([39]; [43]), and empirical research ([38]), meaning making is a plausible mechanism that explains the associations between coping strategies and bereavement-related mental health outcomes for a variety of reasons. Life-altering losses have the capacity to cause a crisis of meaning for the bereaved, in which prior understandings of the world, oneself, and interpersonal relationships are fractured ([43]). Resolving such disruptions in one’s assumptions regarding the world, the self, and relationships caused by the death of a loved one requires a process of meaning reconstruction ([38]). When the death of a loved one occurs suddenly, violently, or includes complex issues of intent and human volition, such as in the case of overdose and suicide loss, a protracted and arduous quest to make sense of the event and its broader implications is common. Indeed, among all the needs (e.g., pragmatic, spiritual, emotional) expressed by those bereaved by such sudden and violent loss, their unmet needs to make sense of the experience and of themselves in light of it are the strongest predictor of their prolonged grief symptomatology ([5]). Moreover, the extent to which bereaved individuals make sense of the loss appears to explain much of the heterogeneity in outcomes within this population ([49]). Clarifying this finding, prior longitudinal research has underscored the mediating role of meaning making in influencing bereavement-related mental health outcomes, specifically illustrating how the degree of meaning made in the first year following loss mediated the relationships between prospective risk factors for prolonged grief disorder (including violent death bereavement) and symptom severity in the context of sudden and violent losses ([36]). Although prior research highlights various coping strategies as correlates of outcomes and supports the facilitative role of meaning making in the context of sudden or violent loss, little is known about how coping strategies impact the meaning making process, and therefore bereavement-related outcomes, in the context of suicide and overdose bereavement.

Noting the limited research on coping strategies in the context of suicide and overdose bereavement and the need to understand the underlying mechanisms that may explain how coping strategies can be differentially adaptive, the current study examined the mediating role of meaning making in the relationships between problem-focused, emotion-focused, and avoidant coping strategies and mental health outcomes. Specifically, this study utilized secondary data analysis to examine the prospective relationships between coping strategies, meaning making, PG, PTS, and depression among a sample of suicide- and overdose-bereaved adults.

## 2. Methods

### 2.1. Procedure and Participants

The current study involved a secondary analysis of data derived from a large online longitudinal study of traumatic loss. The study was approved by the University of Memphis Institutional Review Board (IRB). Criteria for study eligibility included (a) being over the age of 18, (b) having experienced the death of a loved one or someone close to them within the previous five years due to suicide or a fatal opioid-related overdose, and (c) being fluent in the English language. Participants were recruited from a variety of online sources including through posts on various social media sites and organizational listservs relevant to traumatic bereavement. Participants were provided compensation for the completion of each study timepoint.

Data collection for the current study occurred between October 2019 and January 2021 using an online survey. Prospective participants were directed to a website that described the purpose of the study, provided the contact information of the investigators, and provided the option to enroll in the study. Informed consent was obtained from each participant prior to engagement in the study. In total, 300 adult survivors of suicide and overdose completed the baseline measures (T1). A total of 88 respondents who completed the T1 assessment did not complete the T2 assessment six months later. Three declined to participate and the remaining participants did not respond to emailed invitations to continue with the study. The 212 (70.6%) respondents who completed the T2 assessment were included in the current analysis. Of note, there were no significant differences in the study variables between those who were lost at or declined follow-up and respondents who completed the T2 survey.

### 2.2. Measures

#### 2.2.1. Coping Strategies

The brief version of the Coping Orientation to Problems Experienced Inventory (Brief COPE; [9]) is an abbreviated version of the original 60-item COPE ([10]). The Brief COPE consists of 28 self-report items that reflect efforts to respond to serious or stressful life circumstances across 14 broad categories. Although there is some variability in the use of the Brief COPE ([45]), including the factor structure of the instrument, a recent systematic review highlighted that a two-factor solution is the most commonly identified and used: avoidant and approach-based (which includes problem- and emotion-focused strategies) coping ([52]). In addition, prior studies have shown the limited utility of including items that map onto the thematic categories of Humor and Religion ([17]), so these four items were excluded from the current analysis. Example avoidant coping items include “I’ve been telling myself ‘This isn’t real” and “I’ve been turning to work or other activities to take my mind off things”. Example approach-based coping items include “I’ve been taking action to try to make the situation better” and “I’ve been getting emotional support from others”. The items are rated using Likert-type response options ranging from 1 (*I have not been doing this at all*) to 4 (*I have been doing this a lot*). The average scores for each subscale were used in the current analyses. The reliability of each subscale was good (avoidant coping alpha = 0.74; approach-based coping alpha = 0.85) in the current study.

#### 2.2.2. Meaning Making

Meaning making was assessed at T2 using the Integration of Stressful Life Experiences Scale short form (ISLES-SF; [22]), which recent research has confirmed to have an interpretable unidimensional structure ([32]). The ISLES-SF ascertains the degree to which a bereaved respondent has made meaning after a significant loss and uses a Likert-type response scale. Respondents were asked whether they agreed or disagreed with six declarative statements (e.g., “I am perplexed by this loss”; “Since this loss, I don’t know where to go in life”; “This loss has made me less purposeful”). The reliability of the ISLES-SF at T2 was strong, with α = 0.90.

#### 2.2.3. Grief-Related Mental Health Outcomes

The PTSD Checklist from the DSM-5 (PCL-5; [2]) was utilized to measure the severity of PTS symptoms at T1 and T2. The PCL-5 contains 20 items consistent with the DSM-5 diagnostic criteria for PTSD. Following prior research, the suicide or overdose death of a loved served as the index event ([14]). In the current sample, the internal consistency for T1 (*α* = 0.92) and T2 (*α* = 0.94) was strong.

PG symptoms were assessed using the Inventory of Complicated Grief (ICG; [47]), a 19-item measure that assesses yearning or longing for the deceased, a sense of meaninglessness, difficulty with accepting the loss, and identity confusion, among other symptoms. At both T1 and T2, the ICG demonstrated good internal consistency for the current sample, with *α* = 0.88 and 0.89, respectively.

The depressive symptom severity was assessed using the Patient Health Questionnaire-8 (PHQ-8; [30]). The PHQ-8 is an 8-item self-report measure of depressive symptomatology based on the DSM-IV (e.g., “Over the past 2 weeks, how often have you been bothered by: ‘Little interest or pleasure in doing things’ and ‘Poor appetite or overeating’”). For the current sample, the internal consistency was strong for both waves of data collection (T1: *α* = 0.90; T2: *α* = 0.88).

#### 2.2.4. Demographic and Loss-Related Characteristics

Respondents’ age, gender identity, race and ethnicity, and socioeconomic indicators (i.e., education, household income), as well as loss-related characteristics (time since the loss, closeness to the decedent), were obtained at T1. The pre-death closeness between the respondent and decedent was assessed at T1 using the closeness scale of the Quality of Relationships Inventory, Bereavement Version (QRI-B; [6]), which demonstrated strong internal consistency with the current sample, with *α* = 0.87.

### 2.3. Statistical Analyses

Data analysis occurred in two steps. First, using IBM SPSS Statistics (v29), correlation analyses were conducted to assess the relations among study variables and to inform the inclusion of covariates in subsequent mediation analyses using path analysis. Next, after the review of significant correlation coefficients in the first step, we developed and tested a path model illustrating the mediating role of meaning making in the prospective relationship between avoidant coping strategies, approach coping strategies, and grief-related outcomes (PG, PTS, depression symptom severity). Within this path model, which was created using MPLUS 8, avoidant coping strategies and approach-based coping strategies at T1 were covaried with each other, and both were regressed onto T2 meaning making, T2 PTS and T2 PG symptoms, and T2 depression severity. The proposed mediator, meaning making, was regressed on T2 PTS, T2 PG, and T2 depression severity. The model (see Figure 1) was adjusted for all the mental health outcomes at T1, as well as demographic and loss-related characteristics that were significant in the correlation analysis. To promote parsimony, we assessed the model fit statistics for additional models that reduced the number of demographic and loss-related characteristics. To assess the fit across these models, we utilized the comparative fit index (CFI), the root mean square error of approximation (RMSEA), and the standardized root mean square residual (SRMR). CFI values ≥ 0.95 indicate an excellent fit ([25]), RMSEA values ≤ 0.05 indicate an approximate fit ([7]), and SRMR values < 0.08 indicate a good fit ([25]). To assess direct and indirect effects within the proposed model, bias-corrected bootstrapped estimates ([16]) were generated based on 10,000 replications. This procedure yielded a robust test of mediation that could account for departures from the assumption of normality ([20]). The statistical significance of indirect effects was assessed using 95% bias-corrected bootstrapped confidence intervals. Models were estimated using full information maximum likelihood estimation (FIML) to account for missing data (0–4% missing; [34]).

## 3. Results

### 3.1. Sample Characteristics

A total of 212 respondents completed the measures at T1 and T2 and were included in the analyses. The average age of the respondents at T1 was 47.42 years (*SD* = 15.17) and the majority identified as female (*n* = 184; 86.8%) and White (*n* = 170; 80.2%). Regarding the cause of death, 91 participants were survivors of a loss to a fatal overdose (42.9%) and 121 were survivors of a suicide loss (57.1%). On average, the respondents experienced the suicide or overdose death approximately two and a half years prior to T2 (*M* = 29.97 months; *SD* = 18.58). Most respondents reported the death of a child (110; 51.9%), followed by a sibling (35; 16.5%), a partner (20; 9.4%), a parent (14; 6.6%), and a non-nuclear family member (e.g., grandparent, cousin; 14; 6.6%). In addition, 19 respondents reported the death of a close friend, acquaintance, or colleague (8.9%). The respondents reported a high level of closeness with the decedent (*M* = 25.77; *SD* = 4.90; the maximum was 32).

In the total sample and approximately two and a half years after either an overdose or suicide loss, meaning making was moderate (*M* = 17.26; *SD* = 6.15). Recently, research was undertaken to determine a cut-off score that would differentiate those who may be experiencing a crisis of meaning and those who are not ([32]). Using this cut-off score (≥14), 54 (25.6%) respondents were determined to be experiencing a crisis in meaning indicative of a clinically significant impairment in biopsychosocial functioning. In addition, grief-related mental health outcomes were generally elevated in the sample. For instance, the respondents reported high levels of PTS symptoms (*M* = 30.49; *SD* = 17.98). Based on the proposed cut-offs for the PCL-5 identified by a recent systematic review ([19]), 94 (44.5%) met the criteria for a presumptive diagnosis of PTSD. Likewise, PG symptoms were elevated (*M* = 33.18; *SD* = 13.07). Using cut-off scores to identify individuals with probable PGD ([8]), 125 (58.9%) met the presumptive criteria for PGD. Regarding depression, the total scores on the PHQ-9 were also quite high (*M* = 9.36; *SD* = 5.79), with 99 (46.7%) respondents meeting the presumptive criteria for major depression based on recently proposed cut-off scores ([33]). In summary, an average of 2.5 years following their loss, 25% to nearly 60% of the bereaved sample participating in this study continued to show clinically substantial struggles across a range of mental health outcomes. Table 1 presents means and standard deviations of the relevant demographic and outcome variables included in the study.

### 3.2. Bivariate Relations

The initial correlation analyses indicated a variety of bivariate relationships among the study variables (see Table 2). Among the loss-related characteristics, the months since the loss was significantly related to T2 meaning making (r = 0.20, *p* < 0.01) and T2 PG (r = −0.14, *p* < 0.05), T2 PTS (r = −0.20, *p* < 0.01), and T2 depression symptoms (r = −0.17, *p* < 0.05). Greater closeness to the decedent was associated with increased T2 PG symptoms (r = 0.15, *p* < 0.05). The use of avoidant coping strategies at T1 was associated with lower levels of T2 meaning making (r = −0.29, *p* < 0.01) and higher T2 PG (r = 0.41, *p* < 0.01), T2 PTS (r = 0.46, *p* < 0.01), and T2 depression symptoms (r = 0.39, *p* < 0.01). Conversely, the use of approach-based coping strategies at T1 was associated with higher levels of T2 meaning making (r = 0.21, *p* < 0.01) and lower T2 PG (r = −0.14, *p* < 0.05), T2 PTS (r = −0.14, *p* < 0.05), and T2 depression symptoms (r = −0.16, *p* < 0.05). When considered together, these patterns of bivariate relations offered preliminary support for the proposed mediation model.

### 3.3. Hypothesized Model

As shown in Table 3, the hypothesized model provided an excellent fit to the data. Additional models were tested, removing paths for mental health and demographic covariates to improve parsimony. Specifically, the first parsimonious model removed the respondent age and gender covariates, given the weak association between these demographic characteristics and grief-related complications ([35]; [42]), which produced a marginally worse fit to the data. The second parsimonious model was developed based on the bivariate relations identified in the first analytic step of the current study. Similarly, this model produced slightly worse fit statistics for most metrics (SRMR, CFI, χ^2^). The original hypothesized model explained approximately 24% of the variability in T2 meaning making, 64% of the variation in T2 PG symptoms, 52% of the variability in T2 PTS symptoms, and 37% of the variability in T2 depression symptoms. Additionally, path estimates indicated that meaning making at T2 was negatively associated with PG (*β* = −0.66, 95% CI [−0.76, −0.56], *p* < 0.001), PTS (*β* = −0.53, 95% CI [−0.63, −0.43], *p* < 0.001), and depression at T2 (*β* = −0.40, 95% CI [−0.53, −0.27], *p* < 0.001). These results suggest that the original hypothesized model provided a good fit to the data, had notable explanatory power, met the requirements for mediation, and could reliably predict changes in PG, PTS, and depression symptoms. Therefore, the original hypothesized model with all the proposed covariates was retained and is interpreted below (see Table 4 for summary of direct and indirect effects).

#### 3.3.1. Avoidant Coping Strategies

Regarding direct effects, the use of avoidant coping strategies at T1 was negatively associated with meaning making at T2 (*β* = −0.15, 95% CI [−0.29, −0.02], *p* = 0.05) and remained positively associated with PG (*β* = 0.14, 95% CI [0.04, 0.24], *p* = 0.007) despite the partial mediation by the meaning made. In addition, avoidance at T1 displayed a direct effect on PTS symptoms (*β* = 0.14, 95% CI [0.02, 0.26], *p* = 0.02) at T2. In contrast, the direct effect of avoidant coping strategies at T1 on T2 depression symptoms only trended toward significance (*β* = 0.06, 95% CI [−0.07, 0.19], *p* = 0.06).

Regarding indirect effects, the effect of avoidant coping strategies at T1, mediated by T2 meaning making, on T2 PG symptoms (*β* = 0.096, 95% CI [−0.085, 5.31], *p* = 0.05) was significant, as hypothesized. However, the indirect effects of avoidance at T1, as mediated by meaning making, on T2 PTS symptoms (*β* = 0.077, 95% CI [−0.085, 5.70], *p* = 0.054) and T2 depression symptoms (*β* = 0.057, 95% CI [−0.043, 1.34], *p* = 0.061) both failed to reach statistical significance.

In summary, the use of avoidant coping strategies at T1 was directly and prospectively predictive of higher posttraumatic stress and prolonged grief symptoms on average 6 months later, with the latter effect being partially mediated by its deleterious impact on meaning making on the part of participants. However, the use of avoidant coping strategies at T1 only trended toward predicting subsequent levels of depressive symptomatology.

#### 3.3.2. Approach-Based Coping Strategies

Regarding direct effects, the use of approach-based coping strategies at T1 was positively associated with meaning making at T2 (*β* = 0.17, 95% CI [0.04, 0.30], *p* = 0.01). In contrast, we found no evidence of direct effects of the use of approach-based coping strategies at T1 on T2 PG, T2 PTS, or T2 depression symptoms.

However, we found a number of indirect effects of approach-based coping strategies on T2 outcomes. The indirect effects of approach-based coping strategies at T1, mediated by T2 meaning making, on T2 PG symptoms (*β* = −0.113, 95% CI [−4.43, −0.523], *p* < 0.05), T2 PTS symptoms (*β* = −0.090, 95% CI [−4.78, −0.539], *p* < 0.05) and T2 depression symptoms (*β* = −0.068, 95% CI [−1.165, −0.065], *p* < 0.05) were evident, all of which were negative and statistically significant. In summary, the beneficial impact of approach-based coping strategies at T1 on lower levels of PG, PTSD, and depression symptomatology 6 months later was fully accounted for by its prospective prediction of the meaning made of the loss. Figure 2 provides a visualization of these results.

## 4. Discussion

The current study was the first to prospectively examine the relationship between coping strategies and mental health symptoms, as well as the mediating role of meaning making in this process, among suicide- and overdose-bereaved adults. The results advance the understanding of the role of specific coping strategies following traumatic bereavement and a potential mediator that may underlie the association between coping efforts and subsequent mental health outcomes.

After accounting for demographic and loss-related characteristics, as well as the baseline mental health symptom severity, we found a direct and adverse effect of avoidant coping strategies approximately two and a half years following the death on prolonged grief symptoms and posttraumatic stress symptoms assessed six months later. Specifically, a greater reliance on self-distraction, denial, and behavioral disengagement strategies at baseline predicted the worsening of subsequent PG and PTS symptomatology. Furthermore, avoidance predicted greater disruptions in meaning making at T2, which was concurrently associated with reductions in PG, PTS, and depressive symptoms. As hypothesized, the deleterious impact of avoidance-based coping strategies on prolonged grief symptoms was significantly mediated by its effect of disrupting meaning making regarding the loss. This suggests both an adverse impact of avoidance on post-loss outcomes, perhaps caused by precluding adaptive behaviors (e.g., exposure and adaptation to painful reminders of the loss, reorganizing life goals and roles), and its interference with meaning reconstruction following loss. Simply put, it appears that the survivors of suicide and overdose bereavement who coped with traumatic loss through the cognitive and behavioral avoidance of “triggers” had a harder time mastering and making meaning of the experience, resulting in more intense prolonged grief in the months that followed.

This result is consistent with previous research on the association between various forms of avoidance-based coping strategies and bereavement outcomes, including among individuals bereaved by a natural death as well as those bereaved by a sudden or violent death. For example, [3] ([3]) conducted a latent class analysis (LCA) of a large cohort of adults bereaved by various causes of death, identifying low-symptom, high-PGD, and high- and multi-symptom (PGD, PTSD and depression) clusters. He found that depressive avoidance (e.g., of once-pleasurable activities) differentiated the PGD class from the low-symptom group, whereas both depressive avoidance and anxious avoidance (e.g., of reminders of the loss) distinguished the high- and multi-symptomatic group from each of the others. A further study of university students bereaved by sudden or unexpected death documented an interaction between experiential avoidance (of unpleasant thoughts and feelings) and drive sensitivity, such that PGD symptoms were higher for goal-driven grievers reporting more avoidant coping behaviors ([54]). In general, then, attempts to suppress the awareness of uncomfortable emotions or to distract oneself from the problems engendered by bereavement are associated with poorer bereavement outcomes. The current study reinforces the application of this conclusion to suicide and overdose loss survivors and usefully establishes that the use of avoidant coping strategies prospectively and directly predicts both PGD and PTS symptomatology 6 months later, with the former effect being substantially accounted for by its disruption of meaning reconstruction regarding the loss.

In contrast, engagement in approach-based coping strategies at baseline, such as positive reframing, acceptance, seeking informational support, and actively engaging with the problems posed by bereavement, were associated with increased meaning making, which in turn robustly predicted reduced PGD, PTS, and depressive symptoms at the six-month follow-up. Recent research found that among Dutch adults bereaved by various forms of loss, positive reappraisal was predictive of post-loss adaptation, but not when examined against other variables at the multivariate level ([18]). Our results are similar in that the prospective effect of approach-based coping strategies in all three domains of symptomatology was fully mediated by meaning making; once the meaning made was considered, approach-based coping strategies no longer demonstrated a direct effect on the outcomes. This suggests that meaning making may indeed be a central underlying mechanism of post-loss adaptation to traumatic bereavement, in line with previous contemporaneous and longitudinal research ([36]; [49]). Beyond replicating and extending this finding, the current research furthers our understanding of the role of active approach-based coping strategies in facilitating meaning regarding a loss, which in turn predicts improved bereavement outcomes across a range of measures of prolonged grief, posttraumatic stress, and depression.

While the facilitative role of positive appraisal in adaptive meaning reconstruction may be self-evident, other facets of approach-based coping, such as the acceptance of the death and the active seeking of informational support, also likely enhance the meaning making of the loss. Grief and acceptance have been considered opposite sides of the same coin ([46]), and indeed, sense-making and acceptance tend to covary across the first 24 months of loss, increasing as grief symptoms decrease ([23]). The current findings suggest that underlying this association is the greater meaning made of the loss. That is, as acceptance increases, so too does emotional equanimity, with the resulting deliberative processing of the loss permitting its assimilation into mourners’ broader understandings of the world, themselves, and their ongoing relationship with deceased loved ones, thereby reducing intense and prolonged grief reactions ([39]). Furthermore, and particularly in the context of stigmatized losses such as those due to overdose and suicide, information gathering about why and how the person died may help support this meaning making process. Indeed, in a recent study using these same data, informational and meaning needs were rated as more important among the overdose- and suicide-bereaved relative to those who had experienced a sudden natural loss, and in turn these unmet needs predicted greater psychological distress ([5]).

These findings are clearly relevant for psychological interventions in the context of bereavement following an overdose and/or suicide. First, they underscore the deleterious impact of an over-reliance on avoidance strategies for dealing with the range of often profound negative emotions of shock, anger, guilt, yearning, and abandonment clinically observed in the wake of such losses ([27]). What is required is the cultivation of a safe “holding environment” in therapy ([55]) and the enhancement of the client’s window of tolerance ([21]) for such difficult emotions to permit working with them, such as through the practice of mindfulness and other self-regulatory capacities ([44]). With this grounding, traumatically bereaved clients can be coached in safe strategies for approaching the challenges of such a loss, such as through judicious outreach to obtain practical and emotional social support and behavioral engagement with the loss, such as through progressive exposure to reminders of the loved one’s absence from once-shared places and activities ([51]). Moreover, given the evidence that the salutary impact of approach-based coping strategies on grief, trauma, and depression is fully mediated by the meaning making regarding the loss that such coping strategies make possible, developing reflective and narrative procedures for making sense of grief-related experiences and emotions would seem to be a high priority in grief therapy, as a burgeoning body of research suggests ([38]). In effect, exposure-based methods such as the restorative retelling of the traumatic story of the death ([41]), symbolic interactions with the deceased to address the unfinished business that commonly follows suicide and overdose ([24]; [40]), and creative writing procedures to promote the reconstruction of the client’s disrupted sense of self ([12]) directly support the bereaved client’s “effort after meaning” as a major mechanism of change in the context of grief therapy ([39]).

Although the current study was the first of its kind and broadens the knowledge of adaptation to bereavement following overdose and suicide, a number of limitations should be considered. First, the generalization of the findings should be undertaken cautiously given the use of a convenience sample and its composition. Larger samples with more demographically diverse adults would help to address this in future research. Related to this, our study specifically focused on bereavement following suicide and overdose, which limits the generalizability to other forms of bereavement, including bereavement following a natural death. Second, self-report measures were utilized, which introduced the possibility of bias, and future research can leverage clinical interviews to mitigate this concern. Additionally, we examined categories of coping strategies, rather than specific strategies (e.g., thought suppression, cognitive restructuring, acceptance) that could further inform the development of clinical practice and intervention. Similarly, the current study did not examine the role of one’s religious affiliation and religious practices and how religious coping strategies may relate to meaning making and bereavement-related outcomes. Future research taking a multidisciplinary perspective should build upon the findings presented here to provide a more comprehensive understanding of coping strategies, meaning making, and bereavement adaptation. Finally, while we utilized two timepoints, the introduction of further assessments in an extended longitudinal design could further elucidate the nature of the relations among the study variables.

## 5. Conclusions

The heavy psychological impacts of bereavement by traumatic causes including suicide and drug overdose have been well documented. The current study shed light on the role of avoidance- and approach-oriented coping strategies in aggravating or ameliorating the longer-term distress of the survivors of such losses, roughly half of whom showed continued clinical-level elevations of prolonged grief, posttraumatic stress, and depressive symptomatology two years following the death. The results provided cautionary evidence that avoidant coping through denial, distraction, and behavioral disengagement prospectively predicted higher levels of prolonged grief and posttraumatic stress, with the impairment of meaning making about the loss accounting for much of the variance in the former outcome. In contrast, actively approaching others for support and attempting to confront and surmount the problems posed by bereavement consistently predicted a reduction in prolonged grief, posttraumatic stress, and depression symptoms in the months that followed. The latter impacts were found to be fully mediated by the enhancement of meaning making about the loss, carrying practical implications for bereavement support and grief therapy for this vulnerable population of mourners.

## Figures and Tables

**Figure 1 behavsci-15-00671-f001:**
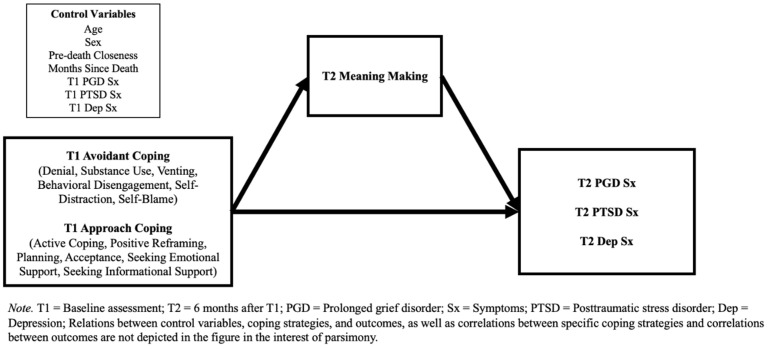
Hypothesized mediation model.

**Figure 2 behavsci-15-00671-f002:**
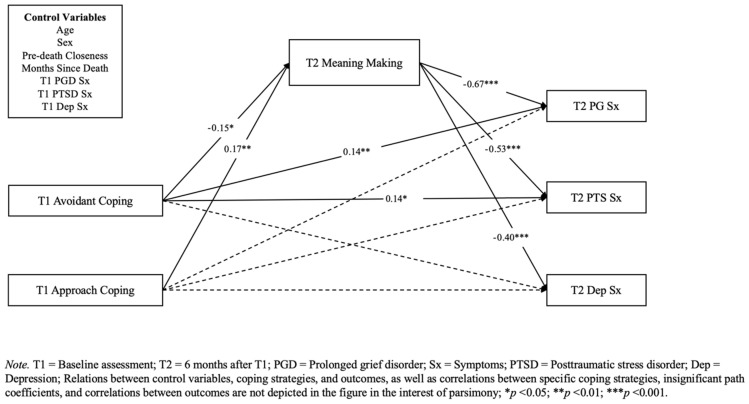
Standardized path coefficients.

**Table 1 behavsci-15-00671-t001:** Characteristics of the sample.

	Total Sample (*N* = 212)		Overdose (*n* = 91)		Suicide (*n* = 121)		*p*
	*M* No.	*SD* %	*M* No.	*SD* %	*M* No.	*SD* %	
Age	47.42	15.17	50.07	13.66	45.97	15.94	0.078
Sex at Birth							0.515
Male	28	13.2	13	14.3	15	12.4	
Female	184	86.8	78	85.7	106	87.6	
Pre-Death Closeness	25.77	4.90	24.84	5.22	26.66	4.46	0.053
Months Since Death	29.97	18.58	28.79	16.21	30.42	19.78	0.043
T2 Meaning Making	17.26	6.15	16.76	5.90	17.59	6.31	0.187
T1 PG Sx	35.75	12.22	36.10	11.23	35.39	13.21	0.300
T2 PG Sx	33.18	13.07	34.18	11.88	32.17	14.25	0.156
T1 PTS Sx	35.23	16.78	34.10	15.66	36.35	17.89	0.162
T2 PTS Sx	30.49	17.98	30.03	17.45	30.96	18.51	0.942
T1 Depression Sx	10.61	6.41	10.71	6.01	10.53	6.69	0.317
T2 Depression Sx	9.36	5.79	9.04	5.52	9.57	5.99	0.573

*Note.* T1 = baseline assessment; T2 = six months after T2; PG = prolonged grief; PTS = posttraumatic stress.

**Table 2 behavsci-15-00671-t002:** Intercorrelations between study variables.

		1	2	3	4	5	6	7	8	9	10	11	12	13
1	Age	--												
2	Sex	0.03	--											
3	MSL	0.26 **	0.01	--										
4	Closeness	0.07	0.03	0.02	--									
5	T1 PG Sx	−0.04	0.08	−0.15 *	0.20 **	--								
6	T1 PTS Sx	−0.1	0.16 *	−0.15 *	0.16 *	0.63 **	--							
7	T1 Depression Sx	−0.09	0.15 *	−0.18 *	0.14	0.58 **	0.67 **	--						
8	T1 Avoidant Coping	−0.13	0.14 *	−0.17 *	0.08	0.52 **	0.62 **	0.54 **	--					
9	T1 Approach Coping	0.03	−0.03	−0.01	0.02	−0.11	−0.26 **	−0.24 **	−0.02	--				
10	T2 Meaning Making	0.05	−0.05	0.20 **	−0.06	−0.40 **	−0.42 **	−0.33 **	−0.29 **	0.21 **	--			
11	T2 PG Sx	−0.12	0.04	−0.14 *	0.15 *	0.50 **	0.46 **	0.36 **	0.41 **	−0.14 *	−0.74 **	--		
12	T2 PTS Sx	−0.12	0.03	−0.20 **	0.11	0.41 **	0.54 **	0.42 **	0.46 **	−0.14 *	−0.64 **	0.78 **	--	
13	T2 Depression Sx	−0.13	0.12	−0.17 *	0.04	0.33 **	0.43 **	0.49 **	0.39 **	−0.16 *	−0.51 **	0.63 **	0.78 **	--

*Note.* T1 = baseline assessment; T2 = six months after T2; MSL = months since the death; PG = prolonged grief; PTS = posttraumatic stress; * *p* < 0.05; ** *p* < 0.01.

**Table 3 behavsci-15-00671-t003:** Fit statistics for hypothesized and alternative parsimonious models.

	Overall Fit Indices					Comparison Fit Indices				
Model	χ^2^	df	CFI	RMSEA (90% CI)	SRMR	Model Comp.	∆χ^2^	∆df	∆CFI	∆RMSEA
M0, Hypothesized Model	16.63	12	0.992	0.046 (0.000, 0.094)	0.023	---	---	---	---	---
M1, Parsimonious Model 1 (No Demographics)	17.54	12	0.990	0.050 (0.000, 0.097)	0.027	M1 vs. M0	−0.91	0	0.002	−0.004
M2, Parsimonious Model 2 (Empirically Informed)	20.49	15	0.990	0.045 (0.000, 0.089)	0.028	M2 vs. M0	−3.86	3	0.002	−0.005

**Table 4 behavsci-15-00671-t004:** Summary of direct effects and indirect effects.

**Outcome**	**PG Symptoms**			
Direct Effects	β	SE	95% CI	*p*-value
Avoidant Coping Strategies at T1	0.14	0.05	0.037, 0.238	0.007
Approach-Based Coping Strategies at T1	0.01	0.05	−0.095, 0.104	0.931
Indirect Effects				
Avoidant Coping Strategies at T1 → Meaning Made at T1	0.10	0.05	0.001, 0.193	0.050
Approach-Based Coping Strategies at T1 → Meaning Made at T2	−0.11	0.05	−0.203, −0.023	0.014
**Outcome**	**PTS Symptoms**			
Direct Effects	β	SE	95% CI	*p*-value
Avoidant Coping Strategies at T1	0.14	0.06	0.022, 0.262	0.021
Approach-Based Coping Strategies at T1	0.04	0.06	−0.085, 0.155	0.570
Indirect Effects				
Avoidant Coping Strategies at T1 → Meaning Made at T2	0.08	0.04	−0.001, 0.154	0.054
Approach-Based Coping Strategies at T1 → Meaning Made at T2	−0.09	0.04	−0.162, −0.018	0.014
**Outcome**	**Depression Symptoms**			
Direct Effects	β	SE	95% CI	*p*-value
Avoidant Coping Strategies at T1	0.06	0.07	−0.068, 0.191	0.061
Approach-Based Coping Strategies at T1	0.01	0.07	−0.141, 0.146	0.972
Indirect Effects				
Avoidant Coping Strategies at T1 → Meaning Made at T2	0.06	0.03	−0.003, 0.117	0.061
Approach-Based Coping Strategies at T1 → Meaning Made at T2	−0.07	0.03	−0.128, −0.007	0.028

*Note.* Significant associations were determined using a 95% bias-corrected bootstrapped confidence interval based on 10,000 replications that did not contain zero.

## Data Availability

The data presented in this study are available on request from the corresponding author due to privacy and IRB regulations.

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
