# Peer review of "Avoidant and Approach-Oriented Coping Strategies, Meaning Making, and Mental Health Among Adults Bereaved by Suicide and Fatal Overdose: A Prospective Path Analysis"

_behavsci, 2025, doi:10.3390/bs15050671_

Round 1
Reviewer 1 Report
Comments and Suggestions for Authors
Title: Avoidant and Approach-oriented Coping Strategies, Meaning Making, and Mental Health Among Adults Bereaved by Suicide and Fatal Overdose: A Prospective Path Analysis
This well-written and interesting manuscript contributes to the literature on coping with loss.
Introduction
- Could the authors provide examples of emotion-focused coping strategies.
- “Based on contemporary bereavement theory (Neimeyer, 2023; Park 2010)” What is meant by contemporary bereavement theory? Could the authors expand on this in the introduction.
Discussion
- How do the findings on coping strategies for adults bereaved by suicide and fatal overdose compare with studies on losses that are not traumatic?
Author Response
Dear Dr. Rapp,
Thank you for the review of our manuscript “Avoidant and Approach-oriented Coping Strategies, Meaning Making, and Mental Health Among Adults Bereaved by Suicide and Fatal Overdose: A Prospective Path Analysis.” We appreciate the time and attention you and the reviewers have given us to improve the manuscript for further consideration for publication in Behavioral Sciences. Your editorial comments and those provided by the reviewers were very helpful in revising the manuscript, and we carefully attended to each of the suggestions as follows:
REVIEWER COMMENTS:
Reviewer #1:
1. Introduction: Could the authors provide examples of emotion-focused coping strategies.
We agree with the reviewer that examples of specific strategies that are characteristic of emotion-focused coping would be helpful. As such, we have now included some example strategies for each of the coping strategy categories we describe early in the introduction.
- Introduction: “Based on contemporary bereavement theory (Neimeyer, 2023; Park 2010)” What is meant by contemporary bereavement theory? Could the authors expand on this in the introduction.
We describe what we mean by contemporary bereavement theory in the sentences that follow “Based on contemporary bereavement theory…”. The contemporary theory we empirically examined in this study is the meaning making theory (meaning reconstruction theory) of bereavement, and we have now made this more explicit in the first sentence under question.
- Discussion: How do the findings on coping strategies for adults bereaved by suicide and fatal overdose compare with studies on losses that are not traumatic?
This is a great question. It is difficult to ascertain true comparisons of the role of coping, meaning, and outcomes between traumatic and non-traumatic bereavement given the heterogeneity of samples used in prior research. Nevertheless, when possible, we have now clarified the composition of the samples we cite from prior research to assist with this inquiry. In addition, we have now highlighted that the purposeful focus on suicide and overdose bereavement in our study limits the generalizability of our findings to other forms of loss, including bereavement following a natural cause of death.
Reviewer 2 Report
Comments and Suggestions for Authors
This is an interesting study, well written, structured and documented. I have three concerns.
It is completely unclear what the authors mean by 'meaning' (1). This could indicate that they are using a minimalist concept, also known as 'thinning', so that should not bother anyone. This vagueness is accompanied by a complete lack of discussion of the role of religious coping (2), as well as any information about the religiosity/spirituality of the participants (3). Perhaps all of this is completely unimportant, but it still requires some explanation. Either way, I think the topic needs an interdisciplinary approach and not just, if that is the intention, a monodisciplinary (cognitive-behavioural?) approach.
Author Response
Reviewer #2:
1. It is completely unclear what the authors mean by 'meaning' (1). This could indicate that they are using a minimalist concept, also known as 'thinning', so that should not bother anyone.
We appreciate the reviewer’s comment about the ambiguity of the term “meaning,” but as this comment is phrased, it is unclear how to proceed with providing an earnest and effective response.
- This vagueness is accompanied by a complete lack of discussion of the role of religious coping (2)
Based on prior studies on the psychometrics of the BRIEF Cope, items reflecting religious coping were not included in the analyses. Given this, the role of religious coping fell outside the scope of the current study but is, as the reviewer points out, a notable construct worthy of investigation. Therefore, we now underscore the absence of religious coping as a limitation of the current study and encourage future research in this area to build upon the current study findings.
- [Also missing] any information about the religiosity/spirituality of the participants (3).
Since religiosity/religious coping was not included as a central construct of this study, we do not include information on the religious characteristics of the sample. We note this now as a limitation of the current study.
- Perhaps all of this is completely unimportant, but it still requires some explanation. Either way, I think the topic needs an interdisciplinary approach and not just, if that is the intention, a monodisciplinary (cognitive-behavioural?) approach.
In the revised limitations section, we now indicate how future research could benefit from a multidisciplinary approach that considers other manifestations of coping, consistent with recommendations of the reviewer.